# Mn-N-C Nanostructure Derived from MnO$_{2-x}$/PANI as Highly Performing Cathode Additive in Li-S Battery

**Xingyuan Gao** [1,2,†]📶, **Ruliang Liu** [1,2,†], **Lixia Wu** [1,†], **Changdi Lai** [1], **Yubin Liang** [1], **Manli Cao** [1], **Jingyu Wang** [1], **Wei Yin** [1,2,*], **Xihong Lu** [3,*] **and Sibudjing Kawi** [4,*]📶

1   School of Chemistry and Material Science, Guangdong University of Education, Guangzhou 510303, China; gaoxingyuan@gdei.edu.cn (X.G.); liuruliang@gdei.edu.cn (R.L.); wlixia@gdei.edu.cn (L.W.); laichangdi@gdei.edu.cn (C.L.); lyubin@gdei.edu.cn (Y.L.); caomanli@gdei.edu.cn (M.C.); wangjingyu@gdei.edu.cn (J.W.)
2   Engineering Technology Development Center of Advanced Materials & Energy Saving and Emission Reduction in Guangdong Colleges and Universities, Guangzhou 510303, China
3   The Key Lab of Low-carbon Chem & Energy Conservation of Guangdong Province, School of Chemistry, Sun Yat-Sen University, Guangzhou 510275, China
4   Department of Chemical and Biomolecular Engineering, National University of Singapore, Singapore 117585, Singapore
*   Correspondence: yinwei@gdei.edu.cn (W.Y.); luxh6@mail.sysu.edu.cn (X.L.); chekawis@nus.edu.sg (S.K.)
†   The authors contribute equally to this work.

**Abstract:** Highly dispersed Mn metallic nanoparticles (15.87 nm on average) on a nitrogen-doped porous carbon matrix were prepared by thermal treatment of MnO$_{2-x}$/polyaniline (PANI), which was derived from the in situ polymerization of aniline monomers initiated by γ-MnO$_2$ nanosheets. Owing to the large surface area (1287 m$^2$/g), abundant active sites, nitrogen dopants and highly dispersed Mn sites on graphitic carbon, an impressive specific capacity of 1319.4 mAh g$^{-1}$ with an admirable rate performance was delivered in a Li-S battery. After 220 cycles at 1 C, 80.6% of the original capacity was retained, exhibiting a good cycling stability.

**Keywords:** Li-S battery; Mn-N-C; polysulfide; polyaniline; MnO$_2$

## 1. Introduction

With the increasing popularity of unmanned aerial vehicles, new energy vehicles and other devices using batteries as the energy source, the demand for batteries with high endurance is increasing. A lithium-sulfur (Li-S) battery is a battery system with sulfur and lithium metal as electrode materials, possessing a high theoretical specific energy (2600 Wh·kg$^{-1}$) and specific capacity (1675 mAh·g$^{-1}$), low manufacturing cost and environmentally friendliness, and is considered a highly efficient energy storage system [1–3]. However, the commercialization of Li-S batteries is still limited by many factors, such as capacity degradation, low Coulombic efficiency, incomplete sulfur utilization and poor cycling life, which result from the dissoluble lithium polysulfide (Li$_2$S$_n$) and poor conductivity of sulfur [4–7].

Many efforts have been devoted to solving these issues by means of (i) utilization of the interlayer, hierarchical structures, surface protection and quasi-solid electrolytes to inhibit Li dendrite formation [8,9]; (ii) adjustment of the formulation of the separator, binder and electrolytes to alleviate polysulfide shuttling [10–14]; (iii) capturing polysulfides by chemical interactions such as bonding of Li with metal compounds, heteroatoms and polymers [15–17]; (iv) confining polysulfides in the matrix of conductive and porous materials [18–25].

Recently, it has been reported that various metals electro-catalytically convert the long-chain polysulfides to short-chain polysulfides and oxidize the insoluble discharge products to sulfur, enhancing the kinetic rate, reducing the overpotential and promoting

reversible redox Li-S reactions [26,27]. Specifically, non-precious metal incorporated with N-doped carbon matrix exhibited an outstanding electro-catalytic performance due to the locally generated active sites during the charge transport and redistribution between N dopants and neighboring transition metals [28,29]. In detail, metallic Lewis acidic sites can form a strong interaction with Lewis basic polysulfide anions; however, Li can bond with electronegative N atoms, accelerating the reversible redox conversion between Li and S by an enhanced adsorption and dissociation of the discharging intermediate products [30,31].

For example, Chunlei song et al. [32] carbonized nitrogen-rich MOF-100 nanosheets in an inert atmosphere to form Co nanoparticles loaded on nitrogen-doped carbon nanosheets through an in situ formation and addition strategy, which was further compounded with carbon nanotubes (CNTs) to generate a three-dimensional network of nanocomposites. This composite cathode could effectively adsorb and catalyze the rapid conversion of polysulfides due to the excellent conductivity and high catalytic activity of Co nanoparticles. Similarly, Yu et al. [33] embedded Co nanoparticles in an N-doped self-supported carbon fabric, facilitating polysulfide conversion and $Li_2S$ oxidation. Without weakening the adsorption, the thermodynamic and kinetic barriers were both reduced for the conversion of polysulfides, resulting in an ultralow capacity degradation rate of 0.034% over 500 cycles.

As another transition metal with good redox properties, however, Mn has not been well investigated in related fields, despite the successful application in $CO_2$ electroreduction and photoreduction [34,35]. In this work, well-dispersed Mn nanoparticles (15.87 nm on average) on N-doped porous carbon were prepared by pyrolysis of a polyaniline (PANI)-coated $MnO_{2-x}$ composite, which was derived from the in situ oxidation polymerization of aniline monomers by $\gamma$-$MnO_2$ nanosheets. The Mn-N-C structure delivered a high specific capacity of 1319.4 mAh·g$^{-1}$ and an impressive cycling stability of 80.6% retention after 220 cycles at 1 C. The highly dispersed Mn nanoparticles, high specific surface area of porous graphitic carbon, abundant active sites and N dopants are believed to be responsible for the good electrochemical performance due to the promoted adsorption, dissociation and fast redox conversion of polysulfides.

## 2. Materials and Methods

### 2.1. Materials

In this work, all chemicals used were commercially available and in analytical grade without further purifications.

### 2.2. Synthesis of MnO$_2$ Nanosheet

Based on a molar ratio of 2:3, the masses of $KMnO_4$ and $MnSO_4 \cdot H_2O$ were 0.316 g and 0.507 g, respectively. $KMnO_4$ solution was added into $MnSO_4$ solution with a pipette (200 μL). After the solution was completely precipitated, brown precipitates were separated from the aqueous solution via three rounds of centrifugation. The obtained precipitates were transferred to a beaker, washed with distilled water and dried at 90 °C for 4 h. After drying, the powder was dissolved and centrifuged followed by transferring the sediment in the centrifuge tube to a beaker, and drying it in an oven at 90 °C for 12 h.

### 2.3. Synthesis of MnO$_{2-x}$/PANI Composite

0.172 g $MnO_2$ was dispersed in 60 mL ultra-pure water, and then 200 μL aniline was added by pipette. An appropriate amount of hydrochloric acid was added to the solution to adjust the pH to about 3. The mixture was transferred to a 100 mL autoclave and heated at 140 °C for 24 h. The black precipitate was obtained by centrifuging and dried in a vacuum oven at 60 °C for 8 h to obtain black powder.

### 2.4. Synthesis of Mn-N-C Structure

The obtained black powder was transferred to a tubular furnace and heated to 900 °C at 5 °C/min under Ar gas for 4 h. The gas flow rate was 60 mL/min.

### 2.5. Material Characterization

The sample microstructure was analyzed by X-ray diffraction (XRD). The patterns were collected at 2θ = 30–70° with a step size of 0.02°/s on a BRUCKER D8 ADVANCE X XRD powder diffractometer, where a Cu target Kα-ray (operating at 40.0 kV and 25.0 mA) was adopted as the source of the X-ray. The morphology was characterized by transmission electron microscopy (TEM) using FEI Tecnai G2 f20 s-twin 200 kV field emission TEM and scanning electron microscopy (SEM) using TESCAN MIRA 3 LMU field emission SEM. The Brunner–Emmet–Teller (BET) area was analyzed by a BELSORP-max automatic specific surface adsorption instrument. The samples were degassed at 300 °C for 2 h to remove the impurities. The carbon nature was characterized by Raman (HORIBA JY LabRAM HR Evolution). The structure of MnOx/PANI was explored using a Bruker Tensor 27 Fourier Transform Infrared Spectrometer with a resolution of 4 cm$^{-1}$. The surface properties of Mn-N-C were analyzed by X-ray photoelectron spectroscopy (XPS) (Thermo Fisher Scientific K-Alpha). Binding energies were referred to the C 1s peak at 284.5 eV.

### 2.6. Li-S Battery Test

The C/S/Mn-N-C electrode was prepared by mixing 10 wt % of Mn-N-C with CNTs and sulfur followed by heating at 155 °C for 3 h. For the test, the C/S/Mn-N-C electrode was punched into a disc with a diameter of 12 mm, which acted as a working electrode combined with Li foil as the reference and counter electrode. The sulfur loading was 1 mg·cm$^{-2}$ on average. 1.0 M LiTFSI in DME/DOL mixed with 1.0 wt % of LiNO$_3$ was adopted as the electrolyte. The galvanostatic charge/discharge (GCD) test was conducted using a NEWARE battery test system between 1.8 V and 2.8 V at various current densities. Cyclic voltammetry (CV) and electrochemical impedance spectroscopy (EIS) tests were conducted using a CS electrochemistry workstation. The scan rate for the CV test was 10 mV/s and the frequency for EIS ranged from 0.1 to 10$^5$ Hz with a 5 mV AC amplitude.

## 3. Results

### 3.1. Synthesis Route

Figure 1 shows the synthesis route where MnO$_2$ nanosheets were first prepared using a precipitation method, followed by in-situ polymerization of aniline to form a MnO$_{2-x}$/PANI composite (the chemical state of Mn$^{4+}$ may change to lower valence states during oxidation polymerization of aniline, thus this is labeled as MnO$_{2-x}$). After pyrolysis in argon gas, a Mn-N-C nanostructure was generated.

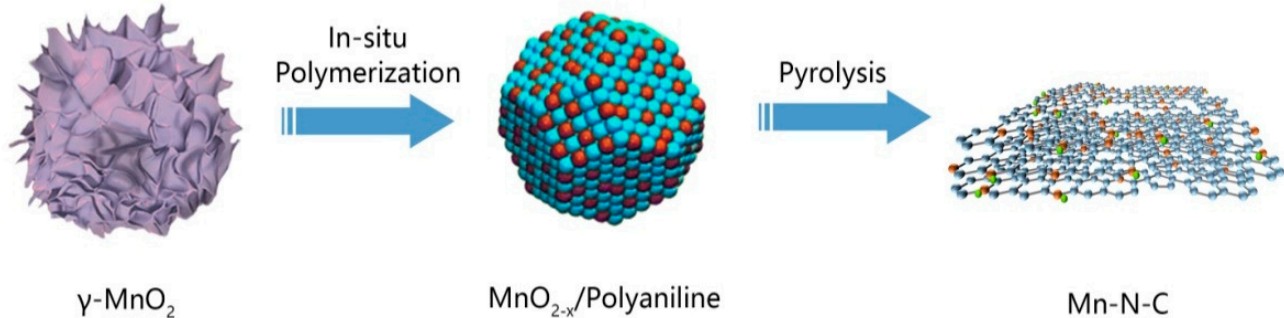

**Figure 1.** Synthetic route of Mn-N-C nanostructure.

### 3.2. MnO₂ Characterizations

As presented in Figure 2a, the crystal structure of $MnO_2$ was γ-phase based on the characteristic peaks of (1 3 1), (3 0 0), (1 6 0) and (4 2 1) at 36.9°, 42.2°, 55.8° and 66.6°, respectively [36–38]. The other very sharp peaks belonged to the silicon substrate. The morphology of the γ-$MnO_2$ is shown in Figure 2b, where $MnO_2$ with a spherical morphology is clearly observed. The diameter of the spheres was 0.53 ± 0.20 μm and the thickness of the nanosheet was 10 nm on average (Figure 2c).

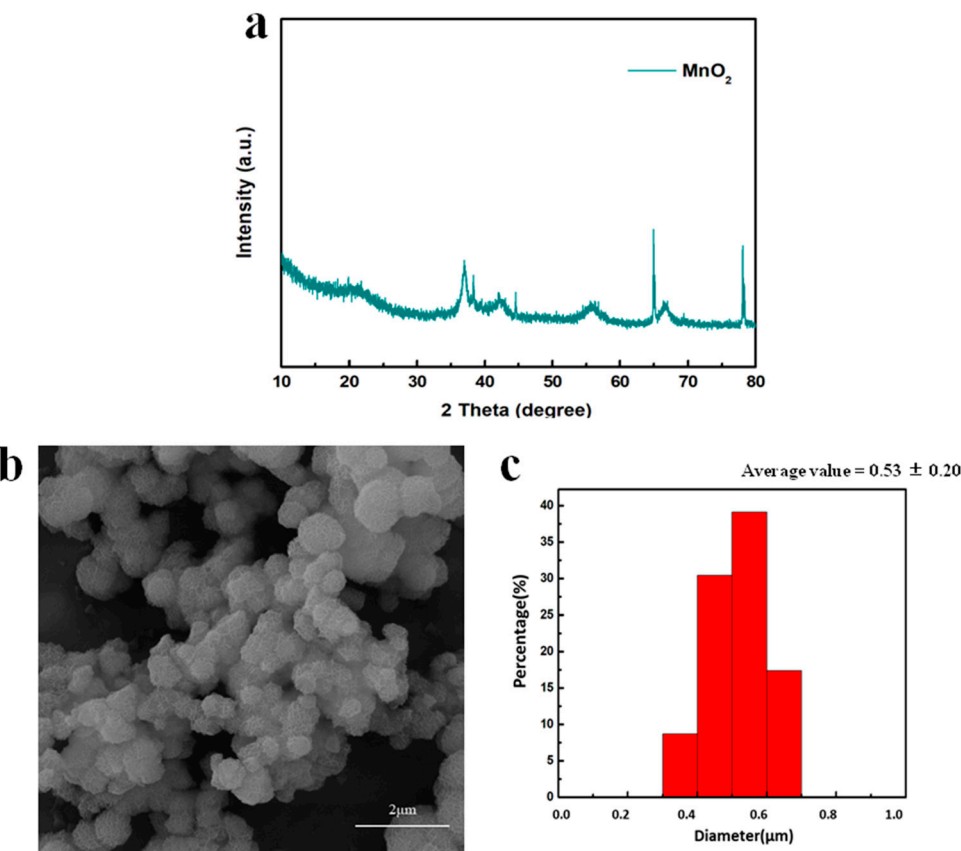

**Figure 2.** (**a**) XRD pattern of $MnO_2$; (**b**) SEM image and (**c**) size distribution of $MnO_2$.

### 3.3. MnO₂-ₓ/PANI Composite Characterizations

After the aniline was added into the $MnO_2$ nanosheets, polymerization of aniline was initiated due to the high redox potential of $MnO_2$ [35,39]. A radical cation was generated during the reaction between the electron lone pair of anilinium cation ions and the metal ion center. After the coupling of radicals, the radical cation of the dimer coupled with the radical cation in aniline, resulting in chain propagation [40]. As shown in Figure 3a, two broad peaks located between 15° and 30° were presented while no obvious $MnO_2$ crystal structures were seen (despite the very broad peak at around 41.9°, which slightly shifted to the left compared with the (3 0 0) crystal phase of γ-$MnO_2$), suggesting the successful formation of amorphous PANI and Mn oxide [35,36,41,42]. As observed in Figure 3b, the nanosheet spherical structures were collapsed into nanoparticles with a diameter of 19.92 ± 2.04 nm, probably attributed to the oxidation polymerization of PANI initiated by $MnO_2$, which changed the ordered crystal structures of γ-$MnO_2$ into a more disordered and amorphous type.

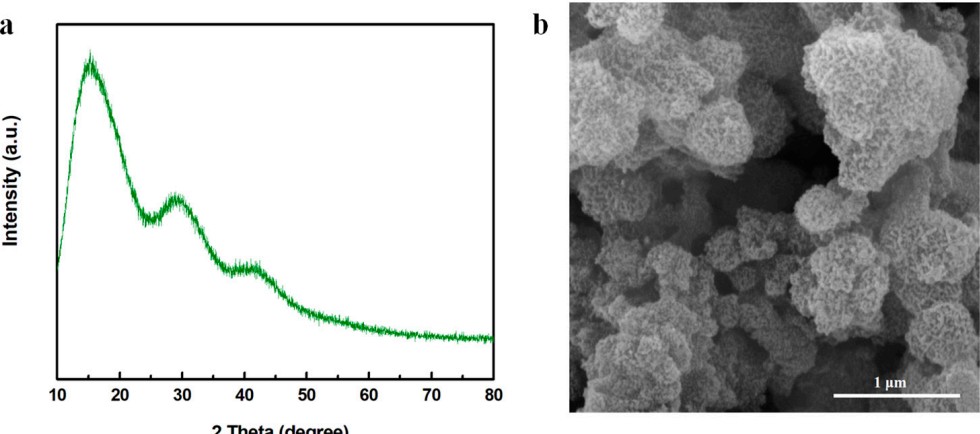

**Figure 3.** (**a**) XRD pattern and (**b**) SEM image of $MnO_{2-x}$/PANI.

FTIR was used to further investigate the structure. In Figure 4a, the peaks at 1504.4 and 1575.8 $cm^{-1}$ were attributed to the stretching vibration of the benzenoid and quinoid ring, respectively. The peak at 1298.1 $cm^{-1}$ corresponded to the Ph-N group and the peak at 1172.7 $cm^{-1}$ suggested the electron delocalization of conductive PANI [43–47]. Moreover, the peak at 503.4 $cm^{-1}$ was assigned to the stretching of the Mn-O bond, indicating the existence of Mn oxides [48,49]. To further explore the surface properties, XPS was adopted and the results in Figure 4b show that O 1s spectra exhibited two peaks at 531.28 eV and 532.94 eV, which referred to surface adsorbed oxygen species and surface adsorbed water molecules, respectively. Specifically, the existence of surface adsorbed oxygen species indicated the formation of oxygen vacancies, which suggested the generation of $Mn^{3+}$ ions in order to compensate the vacancies. In other words, during the polymerization, part of the $Mn^{4+}$ ions in $MnO_2$ were reduced into $Mn^{3+}$ to form a $MnO_{2-x}$ species. This was also consistent with the theoretical assumption that when electrons were transferred from the N in aniline to the Mn ions in $MnO_2$ to form the radical cations to initiate the polymerization reaction, the valence states of Mn should be lowered accordingly. In addition, the absence of lattice oxygen species at a lower binding energy matched the XRD results showing that the $MnO_{2-x}$ was mainly in an amorphous state [37,38]. Besides the O 1s spectra, the N 1s spectra in Figure 4c presented several characteristic peaks at 399.1, 399.8 and 401.2 eV, which corresponded to the quinoid imine, benzenoid amine and [-ph-NH-ph-] groups, in line with the FTIR results showing that PANI was successfully synthesized [36,41]. In summary, due to the high redox potential of $MnO_2$, PANI polymerization was initiated and formed in situ, accompanied by the generation of Mn ions with a lower valence state, resulting in the formation of a $MnO_{2-x}$/PANI nanocomposite.

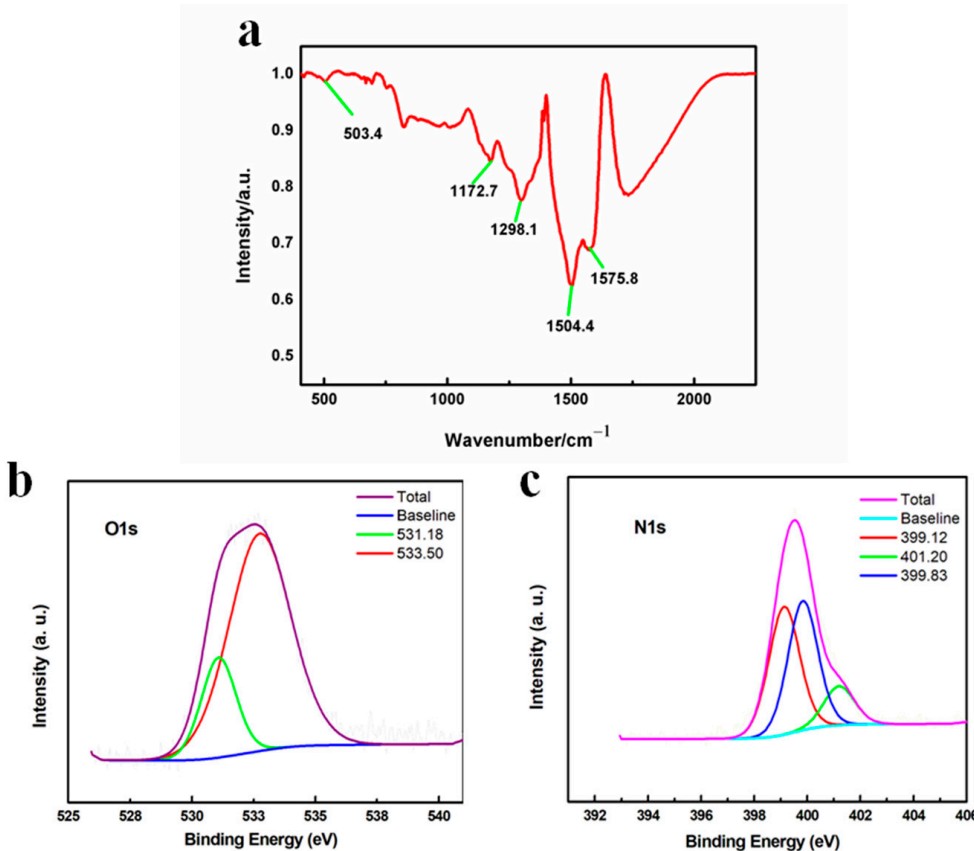

**Figure 4.** (**a**) FTIR spectrum of MnO$_{2-x}$/PANI; XPS spectrum of MnO$_{2-x}$/PANI (**b**) O 1s and (**c**) N 1s.

*3.4. Mn-N-C Characterizations*

The crystal structure of Mn-N-C was analyzed by XRD as shown in Figure 5a. The broad peak at around 25° referred to the (0 0 2) plane of graphitic carbon species, which was generated by the pyrolysis of PANI and enhanced the electrical conductivity [35]. The peaks at 43.16° corresponded to the crystal plane (4 1 1) of metallic Mn [PDF#32-0637], confirming the formation of Mn metals during the thermal treatment of MnO$_{2-x}$ in argon gas. Figure 5b showed that the nanoparticles were highly dispersed on the carbon matrix with a size of 15.87 ± 2.96 nm (very close to 19.92 ± 2.04 nm before calcination), suggesting a strong interaction between Mn ions and electro-donating N in the MnO$_{2-x}$/PANI precursor, which anchored Mn ions and prevented the Mn metals from agglomerating during heat treatment. The slight reduction of the particle size may be attributed to the decomposition of PANI into nitrogen-doped carbon species and the reduction of Mn oxide. As shown in Figure 5c, the lattice space was 0.296 nm, proving the formation of a Mn metallic phase [50]. To further analyze the ordered degree of carbon, the Raman spectrum in Figure 5d demonstrated two peaks at 1591.69 cm$^{-1}$ and 1349.97 cm$^{-1}$, referring to the 2D graphitic carbon nanosheets with C-C stretching vibrations and disordered/defect sp3-hybridized carbon structures, suggesting the transformation of PANI into carbon species. Moreover, the high ratio of I$_G$/I$_D$ (1.02) indicated a dominant amount of more ordered graphitic carbon formation, which promoted the electron transfer [35,51]. In addition, the absence of Mn-O scattering peaks at around 600 cm$^{-1}$ indicated the total conversion of Mn oxide into metallic Mn, in line with XRD and TEM results [35].

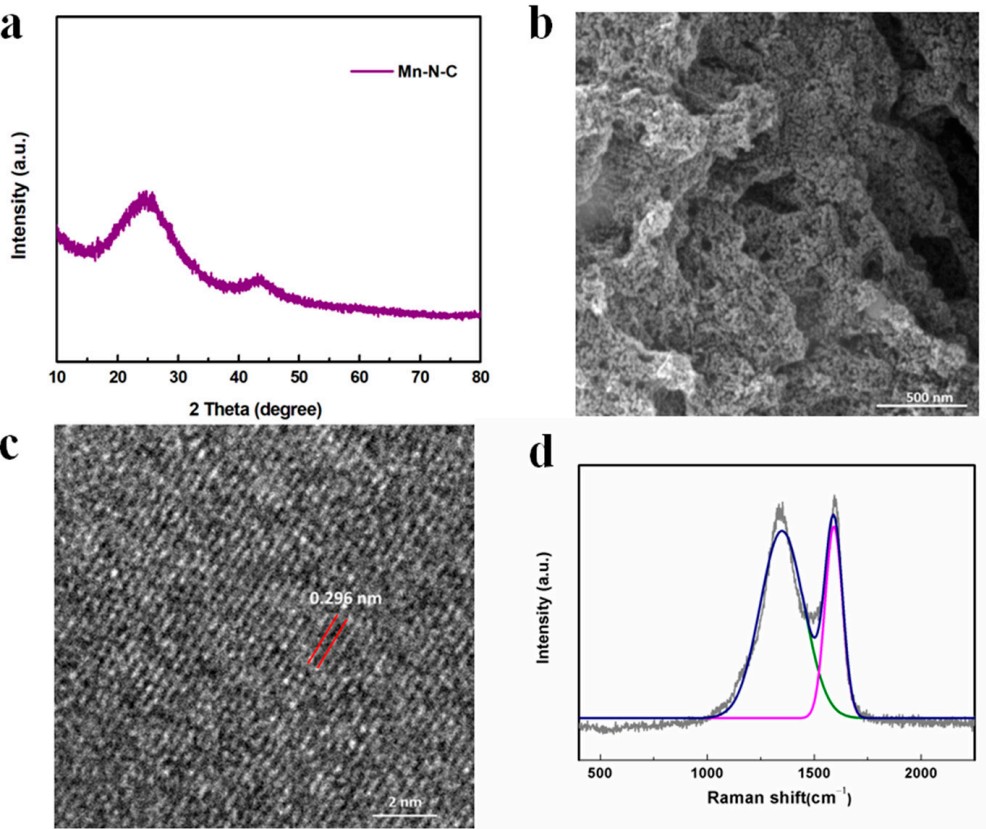

**Figure 5.** (**a**) XRD pattern; (**b**) SEM image; (**c**) HRTEM image; (**d**) Raman spectrum of Mn-N-C nanostructure.

In addition to the electrical conductivity of carbon, which affected the electron transfer, the number of active sites was another determining factor to influence the surface redox reactions, which could be characterized by the specific surface area. Figure 6a exhibits a type I isotherm where a very sharp gas uptake occurred at the low pressure region ($p/p_0 < 0.1$), suggesting that micropores were dominant in the carbon matrix, which facilitated the mass diffusion of electrolyte ions [35,52,53]. In addition, the adsorption and desorption curves almost overlapped, indicating a uniform pore size distribution, which was proven by Figure 6b showing that most of the pores were less than 2 nm (micropores) and the remaining pores were between 2 and 15 nm (mesopores). Moreover, the calculated specific area was as high as 1287 m$^2$/g and the total pore volume was as large as 0.611 cm$^3$/g, offering abundant active sites for the high loading of sulfur, effective adsorption of polysulfides and reversible conversion of polysulfides and other discharging products [33]. The surface groups were analyzed by XPS in Figure 6c,d. In Figure 6c, the peak at 284.5 eV corresponded to the non-oxygenated C-C and C=C bonds [54]. The peak at 285.4 eV indicated the formation of a C-N bond, suggesting the N element was doped in the carbon matrix [55]. The peak at 289.43 eV referred to the C-O species, which could be an impurity [35,54]. As for the N 1s spectrum in Figure 6d, four characteristic peaks were located at 398.5, 399.1, 400.3 and 401.2 eV, corresponding to pyridinic N, Mn-N, pyrrolic N and graphitic N, respectively [35,56]. The existence of graphitic N was consistent with the C 1s spectrum showing that nitrogen atoms were incorporated in the carbon matrix to form a C-N bond and the Raman results showing that highly ordered graphitic carbon species were generated during the pyrolysis of PANI. Moreover, the pyridinic N and pyrrolic N could strongly adsorb metal ions via a coordinated bond, potentially promoting the adsorption of Li ions during the polysulfide conversion [57,58]. In addition, the peak at 399.1 eV referred to the Mn-N bond, which was responsible for the anchoring of Mn atoms by a coordinating mechanism and generating a highly dispersed Mn nanoparticle on the N-C surface [35].

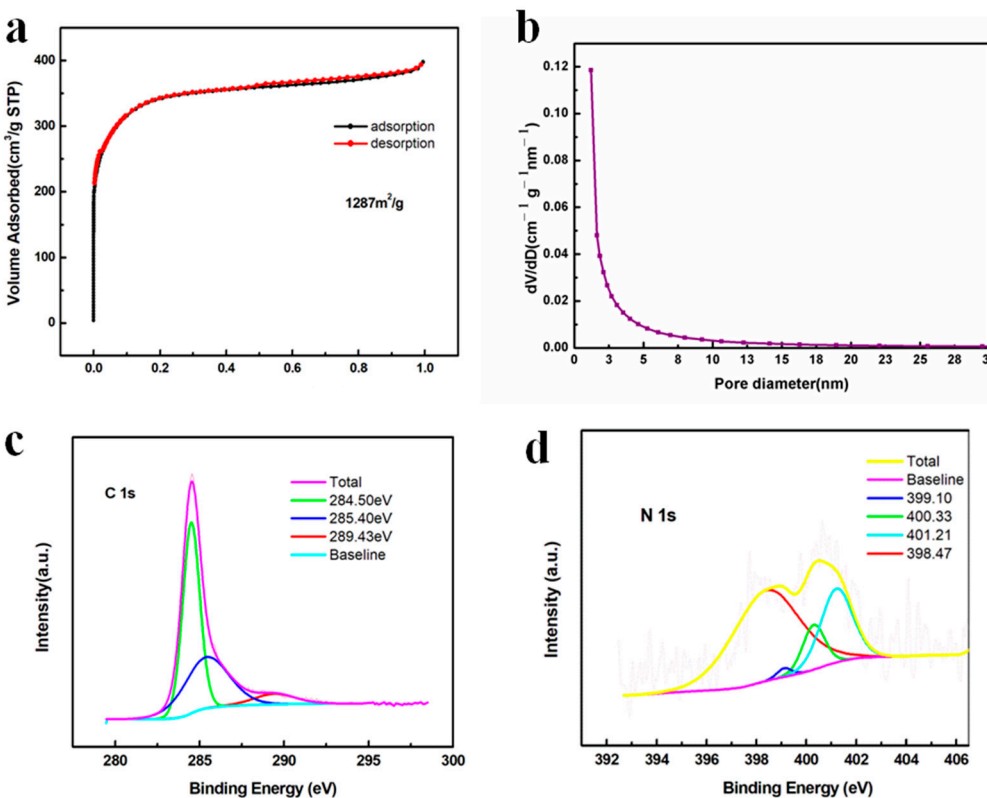

**Figure 6.** (**a**) N$_2$ gas sorption isotherm and (**b**) pore size distribution curve of Mn-N-C; XPS spectra of Mn-N-C (**c**) C 1s and (**d**) N 1s.

### 3.5. Li-S Battery Performances

Li-S batteries were assembled with traditional electrodes (denoted as C/S) and electrodes containing 10 wt % Mn-N-C (denoted as C/S/Mn-N-C), and the rate performance was first investigated under different current densities ranging from 0.1 to 3 C (Figure 7a). At rates of 0.1, 0.2, 0.5, 1.0, 2.0 and 3.0 C, reversible capacities of Li-S batteries using C/S/Mn-N-C as the electrodes were preserved at 1319.4, 1080.2, 922.1, 834.4, 765.0 and 729.8 mAh g$^{-1}$, respectively. When the current density recovered to 0.2 C, the cell still delivered a high reversible capacity of 945.2 mAh g$^{-1}$. In sharp contrast, the capacities of the cell with the C/S electrode ranged 500~700 mAh g$^{-1}$ lower than those of the C/S/Mn-N-C electrode at corresponding rates. The charge-discharge voltage profiles of the cell with the C/S/Mn-N-C electrode exhibited much higher reversible capacities at different rates than those of the pure C/S electrode, and the capacity difference mainly derived from the prolongation of the lower-voltage plateau (2.05 V) (Figure 7b). Specifically, the maximum capacity of 1319.4 mAh g$^{-1}$ achieved at 0.1 C in this work outperformed other carbon cathode materials at similar conditions, such as 1230 mAh g$^{-1}$ at 0.1 C for N,S-co-doped carbon [59], 897.1 mAh g$^{-1}$ at 0.5 C for N-doped carbon nanotubes [60], 1090 mAh g$^{-1}$ at 0.2 mA g$^{-1}$ for acetylene black/S/polypyrrole [61] and 1200 mAh g$^{-1}$ at 167.5 mA g$^{-1}$ for S/C derived from phenolic resin [62]. In addition, the plateau-voltage gap of the C/S/Mn-N-C electrode (150 mV) was smaller than S/C (200 mV). The results indicate that Mn-N-C possessed a better catalytic effect on the conversion of short-chain sulfides (Li$_2$S$_2$/Li$_2$S). The cycling performance of the Li-S cells with different electrodes was also examined at a current density of 0.5 C. The cell with the C/S/Mn-N-C electrode preserved a capacity of 806.3 mAh g$^{-1}$ after 60 cycles, which was much higher than that of the C/S electrode (579 mAh g$^{-1}$) (Figure 7c). Even when the current density was increased to 1 C, the cell with the C/S/Mn-N-C electrode still delivered a maximum capacity of 744.3 mAh g$^{-1}$ with a high-capacity retention ratio of 80.6% after 220 cycles. In comparison, the maximum capacity for the C/S electrode was only 658.5 mAh g$^{-1}$ and the capacity retention ratio was

only 69.1% (Figure 7d). The results of the cyclic performance demonstrated that Mn-N-C could greatly improve the reversible capacity and cyclic stability. The compatibility of electrodes was investigated by electrochemical impedance spectroscopy (EIS) (Figure 7e). When Mn-N-C was introduced into the electrode, the interface impedance slightly increased, indicating the composition was good between Mn-N-C and C/S. Meanwhile, the lower semi-infinite Warburg impedance in the low-frequency regions suggested a faster mass transfer in the Mn-N-C electrode [63]. The reaction kinetics were further studied by the cyclic voltammogram (CV), only one pair of redox peaks appeared with a peak interval of 0.13 V for the cell with the C/S/Mn-N-C electrode (Figure 7f). The CV results with a small interval indicated the Mn-N-C possessed a good reversibility for the conversion of polysulfides.

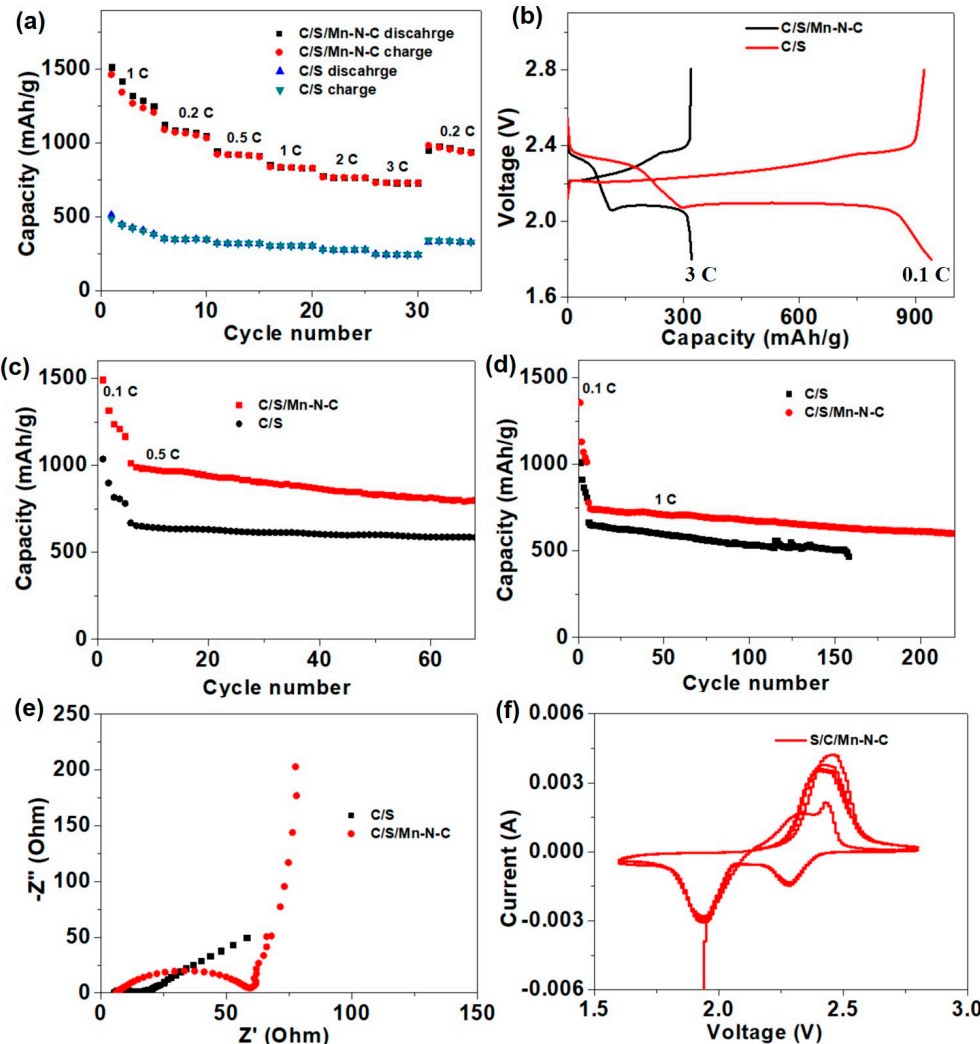

**Figure 7.** (**a**) Rate capacity comparison between Li-S batteries using C/S and C/S/Mn-N-C as the electrodes. (**b**) Galvanostatic charge-discharge voltage profiles of the C/S/Mn-N-C electrode at current densities from 0.1 to 3 C. Cycling performances at (**c**) 0.5 C and (**d**) 1 C for Li-S batteries assembled with C/S and C/S/Mn-N-C electrodes. (**e**) Nyquist plots of Li-S batteries assembled with C/S and C/S/Mn-N-C electrodes. (**f**). CV curves of the Li-S battery using C/S/Mn-N-C as the electrode.

## 4. Conclusions

PANI was synthesized via in situ polymerization of aniline monomers by highly redox $\gamma$-MnO$_2$ nanosheets to form a MnO$_{2-x}$/PANI nanocomposite, which was subsequently transformed into Mn-N-C nanostructures where metallic Mn nanoparticles were highly

dispersed on the nitrogen-doped porous carbon matrix. This nanostructure possessed a large specific surface area of 1287 $m^2$/g and uniformly distributed small Mn metal sites with an average size of 15.87 nm; moreover, a microporous graphitic carbon matrix incorporated with N atoms provided abundant active sites for surface adsorption of Li ions and redox conversion of polysulfides. Owing to the merits above, the Mn-N-C cathode additive delivered an admirable specific capacity of 1319.4 mAh·$g^{-1}$ at 0.1 C and 55.3% retention of the initial capacity after a 30 times current density increase. Moreover, 80.6% capacity retention was achieved after 220 cycles at 1 C, exhibiting a good cycling performance.

**Author Contributions:** X.G.: conceptualization, investigation, resources, writing—original draft. R.L.: conceptualization, investigation, resources, writing—original draft. L.W.: conceptualization, investigation, formal analysis. C.L.: investigation. Y.L.: investigation. L.W.: investigation. M.C.: resources; supervision. J.W.: resources. W.Y.: resource; funding acquisition; supervision; writing —review and editing. S.K.: funding acquisition, resources, project administration, supervision, validation. X.L.: conceptualization; funding acquisition; supervision; writing—review and editing. All authors have read and agreed to the published version of the manuscript.

**Funding:** Generous acknowledgement is given to the MOE (Ministry of Education in Singapore) Tier 2 grant for financial support (WBS: R279-000-544-112), A*STAR (Singapore Agency for Science, Technology and Research) AME IRG grant (No. A1783c0016), NEA (National Environment Agency) in Singapore (WTE-CRP 1501-103), National Natural Science Foundation of China (21802173), Natural Science Foundation of Guangdong Province (2018A030310301), Youth Innovation Talents Project of Guangdong Universities (natural science) (2019KQNCX098) and Characteristic Innovation Projects of Guangdong Universities (2020KTSCX092).

**Institutional Review Board Statement:** No data included in this study involve humans or animals.

**Informed Consent Statement:** No data included in this study involve humans or animals.

**Data Availability Statement:** All data included in this study are available upon the permission from the publishers.

**Acknowledgments:** Generous acknowledgement is given to the MOE (Ministry of Education in Singapore) Tier 2 grant for financial support (WBS: R279-000-544-112), A*STAR (Singapore Agency for Science, Technology and Research) AME IRG grant (No. A1783c0016), NEA (National Environment Agency) in Singapore (WTE-CRP 1501-103), National Natural Science Foundation of China (21802173), Natural Science Foundation of Guangdong Province (2018A030310301), Youth Innovation Talents Project of Guangdong Universities (natural science) (2019KQNCX098) and Characteristic Innovation Projects of Guangdong Universities (2020KTSCX092).

**Conflicts of Interest:** The authors declare no conflict of interest.

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
