# Peer review of "Mn-N-C Nanostructure Derived from MnO2-x/PANI as Highly Performing Cathode Additive in Li-S Battery"

_reactions, doi:10.3390/reactions2030017_

Round 1

Reviewer 1 Report

the paper ranks amongst literally thousands of papers making the ever same things, some highly hirarchically porous graphene, graphite whatever, of course N doped! Decorated with Insitu (in what situation?) Formed nanosized TM X compound which all together has high catalytic activity to catalyse widely unknown reactions. Given they are unknown there is no rigorous evaluation of catalysis. All they do is some cycling where they compare their material with something else which (oh wonder) performs worse than their new wondermaterial. The next PROMISING material is shown. Why they never compare it wit the best previously shown wondermaterial? Insights on which others could build: zero. Impact: zero.

Author Response

Dear Editor Dmitry Yu. Murzin (Prof., Dr.),

Thank you very much for you and the referees’ efforts to the review of our manuscript entitled “Mn-N-C nanostructure derived from MnO2-x/PANI as highly performing cathode additive in Li-S battery”. We acknowledge your and the reviewers’ comments and constructive suggestions very much, which are valuable for improving the quality of our manuscript. After a very serious and careful consideration of the comments, we revised the manuscript as you and referees suggested. Therefore, we sincerely hope you could consider our manuscript for publication in Reactions.

Yours sincerely

  1. Kawi (Ph.D.)

Associate Professor

Department of Chemical & Biomolecular Engineering

National University of Singapore

Singapore

Responses to the editor and reviewers

We thank you and reviewers for your careful review of our manuscript, and really appreciate your constructive comments. Note that all the changes/additions are highlighted in red color in the revised version of the manuscript. Please see below for our detailed responses to the comments.

To Reviewer #1:

Comment: The paper ranks amongst literally thousands of papers making the ever same things, some highly hirarchically porous graphene, graphite whatever, of course N doped! Decorated with Insitu (in what situation?) Formed nanosized TM X compound which all together has high catalytic activity to catalyse widely unknown reactions. Given they are unknown there is no rigorous evaluation of catalysis. All they do is some cycling where they compare their material with something else which (oh wonder) performs worse than their new wondermaterial. The next PROMISING material is shown. Why they never compare it wit the best previously shown wondermaterial? Insights on which others could build: zero. Impact: zero.

Response: Thank you for your comments and valuable suggestions to improve our manuscript. The comparison with similar carbon materials is shown in the context and as below:

Specifically, the maximum capacity of 1319.4 mAh g-1 achieved at 0.1 C in this work outperformed other carbon cathode materials at similar conditions, such as 1230 mAh g-1 at 0.1 C for N,S-co-doped carbon [59], 897.1 mAh g-1 at 0.5 C for N-doped carbon nanotubes [60], 1090 mAh g-1 at 200 mA g-1 for acetylene black/S/polypyrrole [61] and 1200 mAh g-1 at 167.5 mA g-1 for S/C derived from phenolic resin [62].

Reviewer 2 Report

The article reported by Gao describes the S/MnO2 /PANI composite as cathode material for Li-S Batteries. This topic is of general interest. However, as a paper considering the scientific points, some issues need to be considered:

  1. Minor spell check and unit usage corrections are needed.
  2. The charge/discharge curve in Fig. 7b should be written exactly what cycle it came from. Also, "Galvanostatic charge-discharge voltage profiles of the C/S/Mn-N-C electrode at the current density from 0.1 to 3 C." the meaning is unclear and correction is required.
  3. The CV in Fig. 7f was measured only at 10 mV/s, so it cannot be evaluated as a good kinetics for polysulfide conservation. Instead, it shows a similar shape several times in CV, so it should be corrected as "good reversibility."

Author Response

To Reviewer #2:

Comment: The article reported by Gao describes the S/MnO2 /PANI composite as cathode material for Li-S Batteries. This topic is of general interest. However, as a paper considering the scientific points, some issues need to be considered.

Response: Thank you for your positive comments and valuable suggestions to improve our manuscript. The responses are listed as below.

Q1: Minor spell check and unit usage corrections are needed.

Response: Thank you for your comments as well as valuable suggestions to improve our manuscript. A thorough check and corrections have been made accordingly.

Q2: The charge/discharge curve in Fig. 7b should be written exactly what cycle it came from. Also, "Galvanostatic charge-discharge voltage profiles of the C/S/Mn-N-C electrode at the current density from 0.1 to 3 C." the meaning is unclear and correction is required.

Response: Thank you for your valuable comments. We revised Fig. 7b as follows:

Q3: The CV in Fig. 7f was measured only at 10 mV/s, so it cannot be evaluated as a good kinetics for polysulfide conservation. Instead, it shows a similar shape several times in CV, so it should be corrected as "good reversibility."

Response: Thanks a lot for this kind comment. We corrected the discussion about CV results and the corresponding revision in manuscript is as following:

“The CV results with a small interval indicated the Mn-N-C possessed a good reversibility for the conversion of polysulfides”
